# Tuning Alkaline Anion Exchange Membranes through Crosslinking: A Review of Synthetic Strategies and Property Relationships

**DOI:** 10.3390/polym15061534

**Published:** 2023-03-20

**Authors:** Auston L. Clemens, Buddhinie S. Jayathilake, John J. Karnes, Johanna J. Schwartz, Sarah E. Baker, Eric B. Duoss, James S. Oakdale

**Affiliations:** 1Materials Engineering Division, Lawrence Livermore National Laboratory, Livermore, CA 94550, USA; 2Materials Science Division, Lawrence Livermore National Laboratory, Livermore, CA 94550, USA

**Keywords:** anion exchange membranes, crosslinking, electrolysis, fuel cells, alkaline stability, membrane synthesis

## Abstract

Alkaline anion exchange membranes (AAEMs) are an enabling component for next-generation electrochemical devices, including alkaline fuel cells, water and CO_2_ electrolyzers, and flow batteries. While commercial systems, notably fuel cells, have traditionally relied on proton-exchange membranes, hydroxide-ion conducting AAEMs hold promise as a method to reduce cost-per-device by enabling the use of non-platinum group electrodes and cell components. AAEMs have undergone significant material development over the past two decades; however, challenges remain in the areas of durability, water management, high temperature performance, and selectivity. In this review, we survey crosslinking as a tool capable of tuning AAEM properties. While crosslinking implementations vary, they generally result in reduced water uptake and increased transport selectivity and alkaline stability. We survey synthetic methodologies for incorporating crosslinks during AAEM fabrication and highlight necessary precautions for each approach.

## 1. Introduction

There is an urgent need to develop eco-friendly energy conversion and storage technologies that meet global energy demands while reducing greenhouse gas emissions. Renewable energy sources, such as wind and solar, will play a significant role in the future energy mix, and electrochemical technologies such as electrolyzers, fuel cells, batteries, and supercapacitors will be needed to directly address the intermittent availability challenges associated with renewables [1]. Commercial viability and widespread adoption of electrochemical technologies require increases in efficiency and operational lifetimes and decreases in capital expense. Novel material development and advances in cell design can bridge this gap to meet the response times, energy, and power density demands of real-life applications [2]. Electrochemical systems that operate under alkaline conditions are an area of active research that promises capital cost reduction by enabling the use of non-platinum group metal (PGM) catalysts. This review addresses challenges and potential solutions associated with the development of alkaline anion exchange membranes (AAEMs), a class of polymer electrolyte critical to alkaline electrolyte-based electrochemical systems. 

Advances in AAEM technology for energy conversion and storage have been driven by the development of next-generation alkaline fuel cells, alkaline water electrolysis, direct methanol fuel cells (DMFCs), flow batteries, and CO_2_ reduction reactors as outlined in Figure 1. The development of alkaline anion exchange membrane fuel cells (AEMFCs) has been a major thrust over the past 15–20 years, as evidenced by the numerous recent publications and review articles [3,4,5,6,7,8,9,10,11]. Fuel cells generally consist of anodes and cathodes, which are fed by hydrogen and oxygen, respectively. As fuels are oxidized generating power, ions migrate through the polymer electrolyte and unite with the opposite electrode. Proton exchange membrane fuel cells (PEMFCs) are a proven but expensive commercialized technology for light-duty transportation and portable energy applications [12,13]. AEMFCs are envisioned as a lower cost alternative that can operate with non-PGM electrodes and cheaper bipolar plate materials [14]. Various technical and engineering challenges are actively being addressed, including the development of alkaline stable components and PGM-free electrodes with high (>500 mA/cm^2^) reaction rates [5,6]. In 2019, the National Renewable Energy Laboratory (NREL, Golden, CO, USA) set 2024 performance metrics goals for AAEM durability in AEMFCs [15]: chemical stability, ≥1000 h lifetimes with a ≤5% reduction in ion exchange capacity (IEC) in 1 M KOH at 80 °C, and durability, maintain constant open circuit voltage hold for 1000 h at 70% relative humidity at ≥80 °C. Despite recent developments, the vast majority of AAEMs do not meet these stability/performance metrics. As such, identifying new chemistries and materials to meet these metrics is critical. 

Significant efforts are also being applied to develop anion exchange membrane water electrolyzers (AEMWE) [16,17,18,19]. Alkaline water electrolysis currently accounts for about 2–4% of the global production of H_2_ and is an industrial-scale process that has been in operation for nearly 100 years. Alkaline water electrolysis electrode chambers are traditionally separated by a thick, porous diaphragm through which hydrogen constantly diffuses into the oxygen stream [20,21]. As a consequence, industrial-scale alkaline water electrolyzers have low operational capacity and slow response times. AEMWEs can be packaged in compact cells with thin dense membranes that reduce gas crossover, and have large operational capacity and fast response times [19]; however, commercially relevant polymer electrolytes for electrolysis are yet to be realized due to low selectivity and poor alkaline stability [22].

AAEMs may also play a key role in the widespread adoption of methanol fuel cells [23], CO_2_ electrolysis [24,25,26], and redox flow batteries [27,28,29] by promising less expensive and more efficient performance. For example, oxidation kinetics in methanol fuel cells are more favorable in alkaline media [23]. Additionally, CO_2_ electrolysis in alkaline media can convert CO_2_ to ethylene at higher Faradaic efficiencies by minimizing parasitic hydrogen evolution reactions [30,31]. Redox flow batteries are large-scale rechargeable batteries in which the overall capacity is dependent on the amount of the redox couple stored within the electrolyte. Maintaining separate redox components within the anolyte and catholyte is pertinent for long-term energy storage. AAEMs are being explored as solid electrolytes in vanadium redox flow batteries (VRFB) to suppress the crossover of vanadium ions and enhance coulombic efficiency [32]. The lowered vanadium ion crossover in AAEMs is attributed to the Donnan exclusion effect and is reportedly an order of magnitude lower compared to cation exchange membranes. While AAEMs are promising for these applications, significant improvements to stability and mechanical and wear resistance are needed to surpass the current state-of-the-art materials.

Designing AAEMs with long-term alkaline stability remains a significant challenge, particularly for membranes exposed to elevated temperatures or reduced hydration, conditions common within an operating fuel cell [33]. AAEMs are hydrated polymer electrolytes with positively charged functional groups affixed to a polymer backbone that facilitate the transport of hydroxide counter ions. The very design of an AAEM presents a chemical conundrum; hydroxide anions are strong nucleophiles and positively charged cationic moieties are good leaving groups. Currently, the critical bottleneck of operational decline in AAEM-based devices is through hydroxide-induced loss of charge-carrying groups or IEC within the membrane which in turn increases the operating cell voltage. Simultaneously, polymer backbone degradation can lead to increased WU, rapid product crossover, and subsequent device failure. Detailed degradation mechanisms of AAEMs are further discussed in Section 5.1. AAEM development efforts are thus heavily focused on mitigating hydroxide-induced degradation while attempting to achieve and surpass PEM-level conductivities and longevities. An additional, albeit related, challenge to realizing commercially viable AAEM energy conversion and storage technologies concerns the carbonation process that occur when hydroxide ions react with carbon dioxide to form carbonate species. Carbonation reduces hydroxide conductivity over time, which in turn erodes AAEM performance and efficiency [34].

The evolving AAEM landscape has been documented in perspective articles [35,36,37] and several reviews that address important challenges such as water management strategies [38,39], stability issues [40,41], various synthetic approaches [3,42], and molecular modeling [43,44]. Recent developments in AAEM fabrication include designing polymers with sterically protected cations, including embedding cations within the backbone for additional steric protection [45]. Model examples include polybenzimidazole and poly(aryl piperidinium) membranes. These membranes have recently been commercialized under the trade names of Aemion^TM^ and PiperION, respectively. 

While a variety of polymeric systems and charge-conducting functional groups have been explored and advanced, a careful review of the AAEM literature shows that synthesis strategies involving the formation of crosslinks are often employed during membrane fabrication. Crosslinking is also likely prevalent in some grades of commercial AAEMs, including AAEMs from Dioxide Materials (Boca Raton, FL, USA) [46], Xergy Inc. (Harrington, DE, USA) [47], and Ionomr (Vancouver, BC, Canada) [48]. As the first comprehensive review on the topic of crosslinking in AAEMs, we explore how crosslinking is used to tune various membrane properties such as water uptake (WU), mechanical strength, selectivity, alkaline resistance, and resistance to solvent/electrolyte dissolution. This review further describes synthetic strategies for introducing alkaline-stable crosslinkages within AAEMs.

## 2. Crosslinked Ion Exchange Membranes

In polymer science, crosslinking refers to the formation of covalent or ionic bonds that join polymer chains together resulting in an interconnected network. Physically, the formation of crosslinks generally results in increased rigidity and higher glass transition temperatures [49]. The net effect are materials with improved dimensional stability, low creep rates [50], resistance to solvent swelling, and, in many cases, resistance to distortion or softening at elevated temperatures [51]. 

The introduction of crosslinking in AAEMs does not come without its own drawbacks and manufacturing challenges. The most immediate effect is that crosslinked materials are insoluble in all solvents, organic and aqueous. Ion-exchange membranes are often solution cast, but the formation of even a few crosslinks between polymer chains will result in an intractable gel, which can complicate membrane fabrication processes. For this reason, crosslinking is often initiated during the casting step or during post-processing once the desired form factor has been achieved. Crosslinked membranes may also be more complicated to characterize, as their lack of solubility complicates common techniques such as solution-state nuclear magnetic resonance (NMR) spectroscopy and gel permeation chromatography (GPC). Depending on the crosslinking chemistry employed, additional cost of supplies can hinder the economic viability of the membrane. Finally, crosslinked membranes may exhibit lower hydroxide conductivities related to uncrosslinked analogs of similar IEC. However, the increased rigidity imparted by crosslinking counteracts water uptake and allows for mechanically robust (i.e., mechanical strength and swell ratio) membranes at higher IEC. This relationship is discussed in more detail below in this section and in Section 3.2. 

Despite additional processing constraints, crosslinking has been shown to enhance various material properties in both AEMs and PEMs. Perhaps the most well-known ion exchange membrane is Nafion^®^, a perfluoro sulfonic acid PEM developed by Dupont in the late 1960s [52]. Nafion^®^ efficiently transports H+ ions and finds widespread use in the chloro-alkali production of basic commodity chemicals [53] and in PEMFCs. While Nafion^®^ is the gold-standard PEM material, it has some clear limitations such as high cost and poor performance at temperatures above 100 °C, which has motivated the development of alternative PEMs based on polymer backbones such as sulfonated poly ether ether ketone (SPEEK) and phosphonic acid doped polybenzimidazole. Crosslinking design strategies of these comparably inexpensive polymer backbones have overcome several of the limitations of Nafion^®^ by enhancing mechanical stability, and in several cases, enabling proton conductivities that rival Nafion^®^ by improving the material’s water-keeping properties at elevated temperatures and high IECs [54,55,56]. Covalent crosslinking has been shown to mitigate mechanical instability in high temperature phosphoric acid doped PEMs [57,58,59], as well as significantly reducing unwanted methanol crossover in DMFCs compared to Nafion^®^ [60,61]. 

As a general rule, water management is paramount to ion exchange membrane performance. As with PEMs, crosslinking in AAEMs can be used as an approach to mediate the tradeoff of WU with high IECs and other important properties alike. Figure 2 is a graphical illustration of the interconnectedness of several important properties for AAEM development including ion conductivity, WU, selectivity, mechanical strength, and alkaline stability. We have placed crosslinking at the center of this figure because judicious crosslinking can be used to control WU and, in turn, tune the mechanical and ionic properties to meet application-specific needs. This interplay of properties continues to be a challenge for AAEM development and ion exchange membrane development in general. 

When comparing AAEMs to PEMs, both the relative size of hydroxides [62] (including 1–2 additional solvating water molecules [63,64]) and the increased energy barrier for Grotthuss hopping transport [65] necessitates additional measures to counteract the deficient mobility of hydroxides relative to protons, which usually entails increasing the membranes IEC. However, increasing the effective IEC can lead to several consequences: (1) Swelling accompanied by dimensional changes, which can manifest non-uniform mechanical stress related failures in membrane electrode assemblies (MEAs) [66]. (2) Water-dependent weakening is inversely correlated with mechanical strength (e.g., tensile, tear, etc.), leading to membranes that are difficult to handle and are prone to mechanical failure. (3) Excessive WU results in the dilution of charge where the effective ion conductivity becomes that of the intrinsic conductivity of the hydroxide in dilute aqueous solutions [67]. 

On the other hand, excess water loss also leads to poor performance. Reduced local hydration can disrupt the hydroxide’s solvation shell and enhance its nucleophilicity and propensity to attack cationic sites. Dekel and coworkers demonstrated that degradative loss of charged groups in operando follows a sigmoidal decay function that is directly correlated to the dehydration of the membrane [68]. AAEMs are prone to degradation via this negative feedback loop, i.e., the loss of cationic functional groups causes dehydration which in turn causes accelerated loss of additional cation groups. There is usually an optimal range of water that facilitates charge transport while maintaining sufficient mechanical integrity. 

## 3. Effect of Crosslinking on the Material Properties of AAEMs

Effects of crosslinking in AAEMs were evaluated by comparing peer-reviewed articles containing comparative properties (i.e., WU, conductivity, alkaline resistance, etc.) of near-identical AAEM compositions (i.e., polymer backbone chemistry, side chain architecture, cation moiety, IEC, etc.) with and without crosslinking. Many articles varied the degrees of crosslinking and/or crosslinking monomers and were useful for identifying trends; however, we found numerous publications on crosslinked AAEMs that lacked ‘control groups’ (i.e., uncrosslinked AAEMs) and/or lacked quantitative control of crosslinking (i.e., degree or extent of crosslinks), which hindered our ability to derive quantitative trends and assess the beneficial role, if any, imparted by introducing crosslinks. It is also understood that crosslink methodologies would likely not be published if they did not illustrate an added benefit or affect AAEM performance. Additionally, the method of crosslinking and chemical design strategies for enabling crosslinking are discussed further in Section 5. Herein, we attempt to deconvolute some of the dependencies and structure–property relationship upon crosslinking AAEMs, through detailed, variable-specific case-studies.

### 3.1. Crosslinking and Mechanical Properties

Generally, crosslinking provides a physical reinforcement mechanism to counteract plasticization effects of water absorption. The introduction of crosslinks prevents polymer rearrangement, mitigating the characteristic exponential water uptake at high hydration conditions [69]. Significant reductions to both WU and swelling ratios can be realized with even a small equivalence/molar ratios of crosslinkages; representative examples can be found in references [51,70,71,72,73,74,75,76,77,78,79,80]. An exemplary example can be seen in the work by Zhu et al., in which water management was evaluated as a function of crosslinking poly(arylene 6-azaspiro[5.5] undecanium) (PB-ASU), as shown in Figure 3 [81]. Zhu reported that the mechanical properties of uncrosslinked PB-ASU membranes were difficult to test due to mechanical brittleness, even in the hydrated state. Incorporating crosslinkages up to 15 mol% resulted in a remarkable improvement to mechanical properties and was accompanied by nearly 50% reduction in WU and swelling ratio. Perhaps even more remarkable, crosslinked PB-ASU membranes displayed the highest OH^−^ conductivities with a maximum 116.1 mS/cm at 80 °C with an 8.4% crosslink ratio. 

Selectively incorporating monomers with rigidity can further improve mechanical strength. The Zhang group conducted a comparative study by crosslinking poly(ether ketone) with 20 mol% of either flexible alkyl or rigid aromatic crosslinkers, resulting in a 37.7 and 68.9% WU, respectively. Using the rigid crosslinker increased the Young’s modulus by 29% at the expense of decreasing the elongation at break by 30% [82]. Surprisingly, rigid crosslinkers led to greater WU and swell ratios, which the authors hypothesize is related to the ability of more rigid spacers to expand interchain polymer spacing thereby supporting larger, and perhaps more efficient, ion transport channels. This hypothesis was borne out by transmission electron microscopy (TEM) and atomic force microscopy (AFM) analysis and was supported by an observed 75% higher OH^−^ conductivity at 80 °C in membranes with rigid crosslinkers. A similar trend was observed by Zhanhu Guo and team in norbornene-based AAEMs crosslinked with rigid oxydiphenol tether compared to more flexible alkyl tethers of relatively similar lengths [83,84]. We compared material properties of norbonene AAEMs of similar IEC (~2.7 meq/g), specifically, membrane 361 from Ref. [83] containing the rigid oxydiphenol linkage to membrane AEM-2 from Ref. [84] featuring a C_6_H_12_ (hexyl) linkage. While 361 and AEM-1 have similar hydroxide conductivities (~70 mS/cm at 60 °C), 361 displayed 2.7× greater tensile strength 41 MPa vs. 15 Mpa, despite uptaking nearly 4× the amount of water compared to AEM-2 (~72% vs. ~19% at 60 °C). These results highlight how the nature of the crosslinker can be used to tune membrane properties without necessarily impacting conductivity and performance. The differences in WU with rigid versus flexible crosslinker may be crucial for certain applications that require stringent water management. Increased tensile strength, which may also indicate improved toughness, is important for both cell operation and scalable manufacturing.

The hydrophilicity of crosslink tethers provides an additional knob for tuning WU and mechanical strength. Coughlin et al. quantified this effect, finding that more hydrophobic crosslinkers suppressed WU by approximately 50% compared to analogs prepared from more hydrophilic linkages [85]. However, the hydrophilic membranes in this study exhibited higher conductivities (up to 33.7 mS/cm Cl^−^ at 90 °C) at relatively low IEC (<1 mmol/g). The authors’ small-angle X-ray spectroscopy (SAXS) analyses suggest that the higher conductivities of the more hydrated membranes are directly correlated to a more uniform distribution of conducting ions. This study further highlights the interconnectedness of key membrane properties and the role crosslinking can play as a means to tune certain properties. 

In a related study, Liu et al. extended this approach, crosslinking poly(phenylene oxide) polymer (PPO) with polyethylene oxide diamine monomers of varying lengths. Ethylene oxide segments longer than bis(ethylene oxide) resulted in the formation of crystalline domains thereby effectively reducing WU, and increasing the density and strength of the membrane [86]. Excessive crystallinity proved to significantly reduce ion conductivity, and ultimately, bis(ethylene oxide) tethers were found to offer the optimal balance of increased mechanical strength and reduced WU and ion conductivity [86]. In devices where operation at 80 °C is paramount, excessive water saturation can prove detrimental to device performance. To counteract this heat-dependent water absorption, hydrophobic crosslinkers can prove effective in mediating this tradeoff. Liu et al. exhibited only a 10% increase in WU from 30 to 80 °C in thermally crosslinked AAEMs via the formation of perfluorocyclobutane linkages whereas WU increased by ~50% over the same temperature range in non-crosslinked analogs [76].

### 3.2. Crosslinking and (Ionic) Conductivity

Overall, crosslinking was found to decrease the effective ionic conductivity of AAEMs with similar IEC, likely through the reduction in hydrophilic phases, which in turn restricts ionic mobility [76,79,87,88,89]. However, the authors conclude that the relationship is more nuanced, as evidenced in systems that uptake a considerable amount of water, judicious crosslinking is leveraged to reduce charge dilution and improve conductivity [70,76,90,91]. Figure 4 highlights several comparative studies of AAEM systems with and without crosslinking. The nature and extent of crosslinking affects IEC and hydroxide conductivity, Figure 4a. In some cases, crosslinking forms or adds additional quaternary ammonium groups which increases IEC. As noted in Section 3.1, the hydrophobicity of the crosslink can affect WU, which can impact water channel formation and conductivity. Crosslinks add rigidity and increase glassiness of polymeric membranes, and at higher levels, crosslinking can hinder ion exchange resulting in lower measured IEC values. 

We stress that Figure 4 does not capture the highly valuable properties described in Section 3.1, namely WU, swelling ratio, and mechanical strength, that partially guide membrane development. A highly conductive membrane may not be functional if it undergoes severe dimensional changes upon water uptake. Our assessment of the AAEM literature is that there is often an optimal crosslink density (or perhaps lack thereof) to achieve sufficient OH^−^ conductivity, control WU, and maintain mechanical strength, but that this ‘crosslink density’ is both difficult to define and likely polymer/cation/IEC dependent. In other words, there does not appear to be a universal ‘optimal’ crosslink density. 

Crosslink density can be difficult to assess quantitatively. The extent, or degree, of crosslinking, can be determined by applying statistical mechanical models, such as the Flory–Rehner equation, to gel fraction and swell ratio data obtained from solvent swelling experiments [92]. We have not yet found examples within the literature of such studies being performed on AAEMs. As such, the extent of crosslinking is usually reported as a “crosslink ratio” or “degree of crosslinking”, and these values are usually an estimation of concentration reported either as the precent of subunits within the polymer backbone capable of forming crosslinks or the molar equivalence of an added crosslinking agent. 

Nonetheless, many studies have sought to determine an optimal extent of crosslinking. In one such study, Kohl and coauthors looked at crosslinked poly(norbonene) AAEMs and found that crosslink density was indirectly correlated with OH^−^ conductivity at crosslinker concentrations above 5 mol% [93]. Without crosslinking, WU was so high that stable films could not be made because of excessive swelling. Through a series of differential scanning calorimetry (DSC) freezing point measurements, the authors conclude that 5 mol% crosslinker yielded a material with an optimal ratio of bound water molecules per ion pair (N_bound_) and unbound ‘free’ water molecules (N_free_) of 6.00 to 5.21. Some amount of free water is essential for bridging the gap between solvated ion pairs, i.e., the formation of water channels for ion mobility. Increasing the crosslinker to 50 mol% crosslinker resulted in a N_bound_:N_free_ ratio of 7.4 to 0.0, accompanied by substantially lower OH^−^ conductivity. A similar optimization study by Pintauro and team demonstrated a clear improvement in conductivity with crosslinking that coincided with reducing excessive swelling, and an optimal crosslink degree of 8 mol% [94]. Less quantitatively, Moo Lee and team controlled the degree of crosslinking via end-group trimerization by varying the thermal processing conditions at 180 °C. They found 40 min processing time yielded optimal conductivity with a near 50% reduction in water uptake [70].

**Figure 4 polymers-15-01534-f004:**
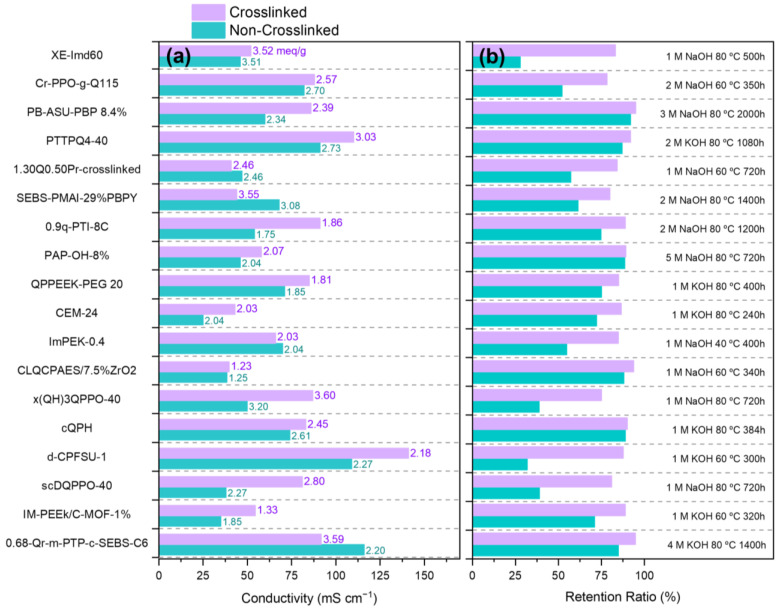
Comparison of recently reported crosslinked (purple) and non-crosslinked (cyan) AAEM analogs. In panel (**a**) bars indicate respective hydroxide conductivities at 60 °C, and the corresponding text labels are measured IEC values. Panel (**b**) shows retention ratios of hydroxide conductivity after exposure to the alkaline stability protocol (black text). XE-Imd60 [70], Cr-PPO-g-Q115 [78], PB-ASU-PBP 8.4% [81], PTTPQ4-40 [95], 1.30Q0.50Pr-crosslinked [96], SEBS-PMAI-29%PBPY [97], 0.9q-PTI-8C [98], PAP-OH-8% [99], QPPEEK-PEG 20 [100], CEM-24 [101], ImPEK-0.4 [102], CLQCPAES/7.5%ZrO2 [103], x(QH)3QPPO-40 [104], cQPH [105], d-CPFSU-1 [106], scDQPPO-40 [107], IM-PEEk/C-MOF-1% [108], 0.68-Qr-m-PTP-c-SEBS-C6 [109].

### 3.3. Crosslinking and Alkaline Stability

We found that alkaline stability within crosslinked AAEMs was often superior to their non-crosslinked counterparts, see Figure 4b, and we assess that this is likely a function of improved water management. Figure 4b extends the hydroxide conductivity data reported in 4a for comparative AAEM systems with and without crosslinking. AAEMs are known to degrade under alkaline conditions through several mechanisms including (1) loss of cation functional groups (via Hoffman elimination and S_N_2 substitution) [110], (2) backbone cleavage [111], (3) oxidation by reactive oxygen species (ROS) generated by nitrogen ylides [112,113,114], (4) polymer dissolution by electrolyzer by-products, and (5) crack propagation. Although experimental and density functional theory (DFT) small molecule studies can be used to survey stability in alkaline conditions [115,116,117], durability and lifetime experiments require extensive screening of the polymer backbone and cations in tandem under relevant environmental conditions (i.e., often while under load in an electrochemical device) to capture all of the aforementioned degradations modes [79,118]. Degradation can be measured in many ways and is often reported as a function of IEC, conductivity, and cell voltage, among other measurements. 

Regardless, several case studies illustrate that alkaline stability is generally improved. Dekel, Willdorf-Cohen, and Hickner et al. showed that a hydroxide’s solvation can significantly influence its nucleophilicity, which is correlated to the rate of AAEM degradation [119,120,121]. Numerous studies have shown that crosslinking results in a larger concentration of bound water molecules, which helps to shield quaternary ammonium groups and acts as a primary mechanism for improving alkaline stability [79,93,122,123,124]. Kim et al. further explored the effect of crosslinker length and crystallinity on water partitioning and alkaline stability, as shown in Figure 5 [86]. The composition of the crosslink was found to have an effect on the ratio of bound water, with BEO (bis-ethylene oxide) resulting in highest ratio of bound-to-free water. Alkaline stability, as assessed by tracking conductivity as a function of age in 1 M KOH at 80 °C, was subsequently shown to coincide with the concentration of bound water molecules. In a similar study, Wei and Ding and coauthors showed that the concentration of bound water increased upon crosslinking of poly(meta-terphenylene alkylene) and attributed the majority of the water becoming trapped in the hydrophilic crosslinked network phases [125]. Alternative crosslinking-induced protective mechanisms may include added steric hinderance imparted by alkyl chain crosslinkages that can shield quaternary ammonium species from hydroxide attack by increasing the energetic barrier for Hoffman elimination [126,127,128]. 

Oxidative degradation arising from the formation of ROS is a major degradation mode for PEMs and is an increasing concern within AAEM literature, including efforts to identify the ROS within electrochemically cycled AAEMs [129,130]. Subsequent studies have shown that increasing the degrees of crosslinking improves oxidative stability toward OH∙ and OOH∙ radicals within AAEMs treated with Fenton’s reagent [131]. Crosslinked AAEMs exhibit a notable reduction in mass loss after exposure to Fenton’s reagent, which is likely a function of network materials requiring more chain scission events to degrade/liberate polymer fragments from the polymer matrix [118,132,133]. This property may imply that crosslinked AAEMs are more resistant to ROS-induced pinhole formation. Finally, we note that increased oxidative stability upon crosslinking is documented in the PEM literature as well [134,135,136,137,138].

### 3.4. Crosslinking and Selectivity

Selectivity refers to the preferential transport of hydroxide ions with respect to the transport of unwanted ions and/or fuel sources [139]. We assess that judicious crosslinking is likely an effective way to increase selectivity. Improvements in selectivity have advanced redox flow battery and fuel cell technology, where maintaining separation between dissimilar ions results in higher Faradaic efficiencies [140,141]. Fuel crossover can also have detrimental effects on the long-term stability and efficiency of these technologies [142].

Among AAEMs, WU is attributed as a major influence on ion selectivity as excessive WU results in the deformation and swelling of the membrane. Logan and co-workers reported that selectivity in AEMs decreases with increasing IEC, suggesting that selectivity is not necessarily a function of fixed charge concentration. Instead, it was concluded that selectivity is inversely proportional to water volume fraction within AAEMs [143,144]. Similarly, conclusions can be drawn from commercial membranes as shown by Yaroslavtsev et al. by comparing conductivity [145]. As discussed in Section 3.1, crosslinking universally reduces WU and thus can be used to increase selectivity within AAEMs. 

A representative example of crosslinking being used to improve selectivity is shown in alkaline DMFCs in which excessive WU with ion exchange membranes can drastically promote methanol crossover [146]. Increasing the extent of crosslinking has proven effective at reducing methanol crossover in DMFCs [147]. Wu et al. demonstrated a near 50% increase in selectivity to methanol via azide-assisted graphene oxide crosslinking within a styrene–isobutylene–styrene (SIBS-hybrid) AAEM [148]. Na and team featured an 88% reduction in methanol permeability by crosslinking polysulfone AAEM [61]. The authors proposed that the underlying mechanism is most likely related to the restriction of hydrophilic channels, which proportionally hinders the diffusion of bulkier ions [87]. 

An additional technology that relies on selectivity is reverse electrodialysis (RED), which is a process for harnessing energy from electrolyte concentration gradients using AAEMs and PEMs in tandem. Crosslinking is a critical tool in RED design used to optimize performance by mediating selectivity [149,150]. Hydrophobicity may also play a critical role affecting the transport of different ions and may be utilized to tune selectivity. Implementing hydrophobic crosslink segments can effectively change the polarity of the local microenvironment and affect the transport of bivalent ions or molecules with large, tightly bound hydration shells as evident in other forms of dialysis applications [151,152,153].

## 4. Application in Electrochemical Reactors

At this stage in the development of AAEM-based technologies, much of the primary focus lies on developing effective non-Pt group catalysts and reactor design. However, electrochemical testing of novel crosslinked AAEM chemistries should proceed in parallel to assist in identifying the most promising strategies for AAEM design. Although many crosslinked AAEMs have been electrochemically tested in rudimentary cell designs, some crosslinked AAEMs have shown relatively promising initial results in hydrogen fuels cells and AEMWEs [70,74,76,83,84,86,88,93,108,109,118,123,125,153,154,155,156,157,158,159,160,161,162,163].

In the pursuit of commercially relevant cell performance, AAEM testing should continue to extend toward more complex reactor designs. To increase the efficiency of fuel cells and CO_2_ electrolyzers, zero-gap reactors are often constructed using gas diffusion electrodes (GDEs) and membrane electrode assemblies (MEAs) [164]. By implementing GDEs, the concentration of fuel species delivered to the cathode is increased, allowing for high current densities. MEAs function to mitigate ohmic losses by hot-pressing AAEMs and electrodes together. Even though hot pressing is the general practice of MEA fabrication, the lack of mechanical stability of AAEMs under high temperature and pressure often limits fabrication to manual compression of hydrated alkaline membranes [165]. Thus, the interfacial conductivity between the catalyst and the membrane is sacrificed in currently available AAEMs used in electrochemical reactors. Despite this downfall, there have been several notable examples of crosslinked AEMFCs achieving power densities > 400 mW/cm^2^ [74,86,88,93,95,123,153,162]. Although an ionomer-based interlayer between the membrane and electrodes can and should be introduced to overcome interfacial limitations due to the absence of hot pressing, it is understood that this approach limits reproducibility and makes comparison across separate research groups difficult to assess. For that reason, all electrochemical testing should include commercially available AAEMs, such as Sustainion, PiperION, among others, to both benchmark and facilitate comparison.

As crosslinked AAEMs have shown superior alkaline stability, research must also include in operando degradation studies. Y. Luo et al. demonstrated the improved in operando durability of crosslinked AAEMs back in 2012. A synthesized AAEM with 5% crosslinking outlasted the non-crosslinked membrane by nearly 3x with only a minimal sacrifice to the peak power density [51]. A similar study was conducted by Young Moo Lee and team where end group trimerization crosslinking resulted in a cell stability test (200 mA/cm^2^) that lasted ~3.2× greater than the uncrosslinked analogs [70]. Both studies demonstrate improved AAEM durability and illustrate the need to continue to contextualize the utility of crosslinking through electrochemical testing as opposed to solely performing ex situ characterization methods, especially in cases where a minor reduction in ion mobility is evident. 

The use of AAEMs in CO_2_ conversion reactors is another emerging application, in part due to the range of products favored by the alkaline environment. Sargent et al. have shown that gas diffusion reactors with AAEMs and a highly alkaline electrolyte enhance CO_2_ to ethylene electroreduction efficiency, higher current densities, and product selectivity in addition to 100 h long operational time [166]. However, a recent review by Berlinguette et al. suggests that the lifetime of CO_2_ reactors is mainly limited by the chemical and mechanical instability of the membrane [26]. Thus, the development of robust AAEM membranes could rapidly advance CO_2_ reactor developments. To our knowledge, crosslinked AAEMs have not yet been explicitly tested in CO_2_ reactors, but they have proven to be efficient and stable separators under similar operating conditions. CO_2_ conversion can produce a multitude of products, both gaseous (e.g., CO, ethylene, etc.) and liquid (e.g., ethanol, formic acid, etc.). Liquid products may alter the electrolyte in deleterious ways, such as resulting in either crossover and membrane degradation through processes such as dissolution. Crosslinking may prove a useful tool for enhancing selectively to reduce crossover or preventing the membrane from being dissolved altogether. Additionally, optimizing the ion selectivity and conductivity of crosslinked AAEMs should permit the development of high-performance CO_2_ reduction reactors that target liquid products such as acetate and formate.

Flow batteries have also demonstrated remarkable improvements to energy efficiency and lifetime through crosslinked AAEMs. Hong et al. demonstrated a crosslinked quaternary ammonium polysulfone membrane that rivaled the performance of Nafion^®^ 115 in vanadium redox flow batteries. Compared to its non-crosslinked counterparts or Nafion^®^ 115, the crosslinked AAEM had higher energy efficiency and no noticeable capacity degradation over 100 cycles [167]. Other studies have shown similar improvements in crosslinked AAEMs over Nafion^®^ as a function of the reduced vanadium permeability [168,169,170].

Continued rigorous steady-state evaluations such as impedance spectroscopy and generating polarization curves should be implemented [171]. In addition to the energetic efficiency and peak power density measurements, long-term stability, accelerated stress tests, and extensive post-mortem analysis are valuable avenues for systematic evaluation of membrane performance [172]. Operando/in situ electrochemical testing of AAEMs are challenging to design but highly encouraged to understand degradation mechanisms and pathways. Lastly, in operando spectroscopic characterization is another approach required to identify pitfalls and degradation modes of AEMFCs, AEMWEs, and CO_2_ electrolyzers [173,174]. Continued efforts from multi-disciplinary teams need to be made in designing, synthesizing, and characterizing materials and evaluating the effect of crosslinking and potential advantages on long-term performance in various AAEM reactor applications as shown in Figure 6.

## 5. AAEM Design and Crosslinking Strategies

### 5.1. Polymer Design

The most common approach for synthesizing AAEMs involves the functionalization of commercially available polymers to add cationic head groups, Figure 7. Among several polymer classes, poly(aryl-ether)s, notably poly(phenylene oxide), polysulfone (PS) and poly(aryl ether ketone) (PEAK) are engineering thermoplastics that have received considerable attention due to their superior mechanical properties, amenability for solution processing, and compatibility with synthetic approaches for introducing cation head groups, tethers, and/or crosslinks to the aromatic component of the backbone [175]. Despite many of these benefits, poly(aryl ether)s struggle with stability towards alkaline electrolytes due in part to the susceptibility of their ether linkages to hydrolysis [176,177]. Polysulfones are particularly susceptible to hydrolysis at both ether and aryl linkages within the polysulfone backbone due to the electron withdrawing nature of the sulfone functional group [111]. In addition, these sulfone linkages exert long-range inductive effects on nearby benzylic carbons, increasing their susceptibility to hydroxide attack [178]. 

Deficiencies of poly(aryl ether)s have motivated the pursuit and synthesis of other ether-less hydrocarbon backbones such as polystyrene [179], polynorbornene [180], polyethylene [181], and ethylene tetrafluoroethylene (ETFE) [182] as shown in Figure 7. These polymer systems have demonstrated good performance and are increasingly attractive especially as evidence mounts of poly(aryl ether) containing AAEMs suffering from deleterious reactions, including the loss of specific functional groups or backbone chain scission [77,78,130]. Finally, poly(aryl piperidium) [183,184] and poly(benzimidazolium) [185] AAEMs have shown immense promise for enhanced longevity under alkaline conditions, which is likely due to increased hydroxide resistance provided by steric hinderance (i.e., protection) of ammonium groups within their respective polymer backbones.

Several nitrogen-based cationic functionalities have been explored, and to date, trimethylamine (TMA), imidazolium, piperidinium, and other cyclic amines comprise the majority of recent AAEM research (Figure 7). Other cations, such as ruthenium metal complexes [186], phosphonium [187], sulfonium [188], and guanidium [189], have also been evaluated. As mentioned, several small molecule surrogate studies have been developed to study alkaline resistance of the head group and to determine the mechanism of degradation, see Figure 8. Generally, quaternary amines commonly decompose via Hoffman elimination, S_N_2 substitutions, or direct nucleophilic attack [111]. Cyclic ammonium cations can reduce Hoffman elimination due to their cyclic conformation and studies have shown significant reductions in the amount of Hoffman and/or demethylation degradation products in piperidinium containing AAEMs [115]. Imidazolium ions are susceptible to nucleophilic attack at the C2 position and addition of bulky groups at the C2 position has shown to enhance imidazolium resistance toward alkaline degradation [115,190,191,192]. Benzylic positioning of cation moieties are more susceptible to degradation due to the formation of ylides. Ylides catalyze the formation of ROS and hydroxyl free radicals, as evidenced by the increased rates of IEC degradation under oxygen versus nitrogen atmospheres [112]. 

### 5.2. Crosslink Chemistry

The implementation of crosslinking requires the careful selection of the appropriate chemistry. Ideal crosslink chemistries should be resistant to hydroxide to facilitate (or not impede) the formation of hydrophilic phase segregation while not hindering hydroxide transport. Metal catalysts used to implement crosslinking may need to be removed or shown to be inert during electrochemical operation. 

Crosslinking preparation can generally proceed in one of two ways, which we refer to as either a “cast and cure” or an ”in situ” approach to crosslinking, as shown in Figure 9. Traditional cast and cure methods (Figure 9a) first require synthetic functionalization of linear chain polymers. The functionalized polymer solution is mixed with additional functional monomers/catalyst and casted directly onto a glass slide. The crosslinking of the polymer strands is then initiated after the film has been casted. In situ crosslinking (Figure 9b) entails the polymerization of a resin formulation into a freestanding crosslinked network in one cohesive step. In both methods, cation functionality can be added before or after crosslinking/film formation. Both cast and cure and in situ describe solution processing of membranes. Alternatively, electromagnetic irradiation of dried membranes via exposure to an electron beam or gamma source may also be deployed to induce crosslinking. The remainder of this section will review various crosslinking approaches and summarize the synthetic steps, necessary considerations, and characterization/testing of the resulting membranes. 

#### 5.2.1. Poly-Functional Amines

The addition of polyfunctional amines is a very common and practical method to crosslink AAEMs via the cast and cure approach, as shown in Figure 10. Advantages include the ease of manufacturing and the availability of a large array of off-the-shelf, low-cost polyamine candidates, as well as the fact that quaternary amine cationic head groups needed for AAEM activity are made directly during crosslinking. 

Tertiary diamines can directly form quaternary amine crosslinkages via Menshutkin-type reactions. The length of the tether, i.e., the molecular spacer between nitrogen groups, can have a significant effect on the extent of diamine quaternization, with C > 2 spacers generally preferred to C1 spacers, as seen in systems utilizing alkyl diamines of various lengths (**1**) [147,193]. In a recent study by Aili, Yang, and coauthors, shorter diamine tethers, such as a systems comparing 4,4-trimethylene-bis(1-methylpiperidine) (**2**) and 1,4-bis(2-methylimidazol-1-yl)butane (**3**), were found to have higher alkaline stability but lower conductivity on account of reduced WU [118]. Multifunctional cationic tethers have also been employed to increase IEC and reportedly result in more flexible and ductile membranes (**4**), possibly due to enhanced WU [104,163,194]. Alternatively, AAEM crosslinking methods can be designed in the absence of a difunctional crosslinker, and instead can be cast as cohesive films with both polymer-bound tertiary amines and halides, which simultaneously undergo Menshutkin quaternization. These procedures are then followed up by TMA solution quaternization to form cations [109,123,154].

Commonly, crosslinking of tertiary amines via the Menshutkin reaction often coincides with drastic increases in IEC due to the simultaneous formation of additional cation groups, which can result in increased WU leading to decreased chemical and mechanical stability [167]. To avoid this potential tradeoff, primary amine crosslinkers, such as 4,4′-diaminobenzophenone (**5**), have also been successfully incorporated [170,195].

#### 5.2.2. Radical Polymerization

Free radical polymerization has been used for AAEM preparation via both traditional cast and cure (Figure 9a) and in situ approaches (Figure 9b), and generally relies on the presence of alkene functionality, such as styrenes, acrylates, isoprene, or olefins. The nature of the alkene group often dictates the processing temperature. Acrylate systems are polymerizable at or slightly above room temperature, whereas unsaturated double bonds of isoprene-based AAEMs (**6**) require more rigorous temperatures (>100 °C) [196]. 

Radical-mediated crosslinking via the cast and cure method via polymer backbones functionalized with acrylate and/or styrene functional groups, such as those containing vinyl benzyl trimethylammonium (**7**) or 2-(dimethylamino)ethyl methacrylate (**8**), and has generally proven to be an effective and straightforward fabrication strategy, similarly depicted in Figure 11 [80,91,197]. One advantage of the in situ free radical approach is that it can be initiated via actinic radiation, which can enable advanced manufacturing methodologies such as additive manufacturing (AM). Yan et al. demonstrated a straightforward method to fabricate AAEMs using UV light to polymerize a resin consisting of styrene, acrylonitrile, and either vinyl imidazolium or vinyl benzyl TMA ionic liquids, and ~4 wt% of divinyl benzene (**9**) as a crosslinker [198,199,200]. Hickner and team manufactured similar AAEMs using triallyl methyl ammonium iodine (**10**) as a multifunctional ionic conducting crosslinkages [158]. Hydroxide conductivities of these direct membrane polymerized AAEMs yielded values of 55.8 mS/cm at 60 °C, and 51.3 mS/cm at 80 °C, respectively [158,200]. Similarly, Hickner et al. fabricated and profiled AAEMs via a stereolithography approach using similar UV-sensitive resins [201]. High crosslink densities were required to enable spatially resolved curing necessary for 3D printing and were achieved using a resin containing 70–80% crosslinkers polyethylene glycol diacrylate, diurethane diacrylates, and vinyl benzyl chloride (VBC) (**11**). 

#### 5.2.3. Olefin Metathesis

Olefin metathesis has enabled the synthesis of AAEMs with potentially more alkaline resistant ether-less polymer backbones via both cast and cure and in situ methods (Figure 12). This reaction involves the scission and regeneration of carbon–carbon double bonds and is often catalyzed by transition metal carbene complexes. Ruthenium-based Grubb’s catalysts are most commonly used for AAEM synthesis due to their compatibility with a wide range of reaction conditions (i.e., tolerance towards a variety of solvents and functional groups) [202]. An early example of direct membrane polymerization via olefin metathesis involved the copolymerization of tetraalkylammonium-functionalized norbornene (**12**) with dicyclopentadiene (**13**), which yielded a mechanically robust AAEM with 28 mS/cm hydroxide conductivity at 50 °C [203]. A single monomer system based on 2,2′-(hexane-1,6-diyl)bis(2-methyl-2,3,3a,4,7,7a-hexahydro-1H-4,7-methanoisoindol-2-ium) iodide (**14**) resulted in similar conductivities; however, the resulting membrane was found to be highly susceptible to degradation via Hofmann elimination [204]. Alternative crosslink tethers such as (3aR,4S,7R,7aS)-2-methyl-2-(3-(trimethylammonio)propyl)-2,3,3a,4,7,7a-hexahydro-1H-4,7-methanoisoindol-2-ium iodide (**15**) may mitigate this degradation pathway [83].

A recent article by the Coates laboratory highlighted the importance of tuning the ratio of the crosslinker to chain-extender in a system consisting of difunctional ammonium monomer (**16**) with cyclooctene (**17**) [205]. A ratio of 1.5:1 **17:16** yielded a maximum conductivity of 69 mS/cm (σ_OH−_ at 22 °C), while increasing the concentration of **16** eroded the overall mechanical integrity of the membrane to the extent that conductivity measurements were no longer possible. Likewise, increasing **17** diminished conductivity, likely due to decreased IEC within the resulting membrane. This work further highlighted the role of catalyst loading on crosslink density. Reduced catalyst loading generally resulted in higher ion conductivities and higher WU as result of lower crosslinking densities, albeit at the expense of mechanical strength. In subsequent work, Coates and team found that 4,5-subsituted imidazolium octene monomers (**18**) reacted with **17** to form materials composed of macrocyclic oligomers instead of high molecular weight polymers [206]. The resultant AAEMs showed high conductivities of 59 mS/cm (σ_OH−_ at 50 °C) and excellent alkaline stabilities. The proposed mechanism leading to macrocyclic formation may be related to the moderate-to-low ring strain and planar feature of the bicyclic imidazolium fused octene that slows propagation rates and acts as a “U-turn” halting further propagation. 

Cast and cure crosslinking via terminal end group metathesis has also been implemented in AAEM synthesis [207]. Li et al. performed ring closing metathesis of a pendant sterically protected charged imidazolium group 2-mesityl-4,5-diphenyl-1-nonene-imidazole (**19**) [208]. Hickner et al. investigated the ROMP-crosslinking of a RAFT-derived polystyrene containing 1-(but-3-enyl)-4-vinylbenzene (**20**) using a second-generation Grubb’s catalyst and showed the order of quaternization, i.e., before or after olefin metathesis-mediated crosslinking, had significant effect on AAEM phase segregation [209]. In the presence of quaternary amines, the Grubbs catalyst was found to have lower catalytic activity and a slowed reaction rate. Subsequent SAXS analysis confirmed an increase in lamellar features in pre-quaternized samples likely due to the increase time afforded to the growing cationic-functionalized polymer chains to restructure and relax during curing. 

The mechanical and chemical stability of ROMP-AAEMs can be improved through the hydrogenation of unsaturated bonds [206,210]. Polynorbornene polymers can become crystalline after hydrogenation [211], which may have a similar mechanical reinforcing effect to that observed in Nafions^®^ PTFE crystalline regions. Hydrogenation can be carried out using low loadings of Crabtree’s catalyst added during the casting step under a hydrogen atmosphere [212]. Although olefin metathesis presents a lot of promise, the implementation of crosslinkable methodologies often requires the detailed design and synthesis of complex difunctional monomers.

#### 5.2.4. Organic Azide-Mediated Crosslinking

The unique reactivity of the organic azide group has found many uses for fabrication of AAEMs. Azide functionalization is relatively easy and is often accomplished via nucleophilic substitution with sodium azide. Organic azides can form reactive nitrenes at elevated temperatures or can be brought to selectively react with alkynes to form triazole-linkages as shown in Figure 13. For the latter, a common approach is the copper(I)-catalyzed azide–alkyne cycloaddition (CuAAC) click reaction, which attaches amine functionalities to azide-decorated polymers [213,214]. The resulting triazole ring can even be utilized as a cationic head group following N-alkylation to generate a triazolium ring; however, in one such attempt, the resulting hydroxide conductivity was found to be fairly low (5–15 mS/cm between 20 and 80 °C) even after a long period of ion exchange [215,216]. The authors hypothesize that triazolium rings may impart strong columbic interactions with nearby hydroxide ions resulting in reduced transport. 

Difunctional monomers, i.e., dialkynes or diazides, can be used as a crosslinking agent via the cast and cure approach. As an example, Liu et al. utilized various dialkynes (**21**) to crosslink ammonium fluorene-based polysulfone (AMPFSU) using 10 mol% copper(I) bromide during the casting process at 120 °C [106]. One issue with CuAAC is that additional steps often need to be taken to remove excess copper catalyst as it has been identified to promote the formation of oxidative free radicals within PEM literature [217]. In the case of Liu et al., this was accomplished by soaking the resulting CuAAC membrane in dilute ammonium solution to oxidize and remove copper. Although copper(I) salts are excellent catalysts for azide–alkyne cycloadditions, this reaction can be brought about in the presence of other catalysts or even in the absence of catalyst altogether at elevated temperatures [218]. Xue et al. employed transition metal-free tetraalkylammonium hydroxide as a catalyst to crosslink quaternized PS with PPO using diethynyl benzene DEB (**22**) to form a semi-interpenetrating network AAEM (semi-IPN AAEM) via the cast and cure approach at 70 °C [219]. Subsequent spectroscopic analysis of the semi-IPN AAEM revealed remaining azide functionality suggesting that the crosslinking reaction was incomplete. Fortunately, unreacted organic azides are not suspected to hinder membrane performance under load in electrochemical devices [213,219].

Organic azides can form reactive nitrene intermediates at temperatures above 130 °C or with UV light. Nitrenes can then proceed to react with aromatic and terminal olefins in the absence of oxygen [220], as depicted in Figure 14. Wang, Li, and coauthors demonstrated crosslinking of AEM scaffolds consisting of PPO [75] or polystyrene [221] polymers equipped with benzylic azides (**23**) at 135 °C for 18 h. In a similar approach, small-molecule polyazide reagents can be added to polymer casting solutions to initiate random crosslinking, as demonstrated by Kim et al., who used 2,6-bis(4-azidobenzylidene)-4-methyl-cyclohexanone (**24**) to form terminally crosslinked poly(arylene ether sulfone) via a 180 °C thermal post-processing of dried membrane films [79]. UV irradiation may serve as a gentler approach to form nitrenes, and as an example, the Li group formed a crosslinked semi-IPN AEM from PPO and quaternized-PS directly via casting from chloroform with exposure to 365 nm UV (250 W) over approximately 3 h [89]. Finally, the relatively indiscriminate reactivity of nitrenes can be a potentially useful feature. Lü et al. incorporated reduced graphene (rGO) into quaternized azide-functionalized polyether sulfone (QPSU) through a post heat treatment step at 160 °C for 24 h [132]. The incorporation of rGO helped control WU, reduced methanol permeability by two orders of magnitude, and enhanced membrane resistance towards ROS.

#### 5.2.5. Thiol-Ene Click Crosslinking

Radical initiated thiol-ene click chemistry entails the reaction of thiols with alkenes to form thioether linkages [222], as shown in Figure 15. Thiol-ene has been used to produce crosslinked AAAEMEMs via both cast and cure and in situ approaches. Similar to radical polymerization described in Section 5.2.2, the thiol-ene reaction can be initiated thermally or photolytically. The resulting thioether bonds are reportedly relatively inert and have been identified as only a minor degradation pathway within AAEMs under alkaline conditions [223]. Dithiols are commercially available and have been utilized to crosslink both PI-ran-P[VBTMA][Cl] [85], and PPO systems via photo-initiation using alkyl dithiol molecules (**25**) [223]. Tibbits et al. demonstrated an in situ method to rapidly fabricate AAEMs via photocuring polyfunctional thiols with vinyl-functionalized ammonium and/or imidazolium monomers such as diallyldimethylammonium chloride (**26**) and 1,3,5-triallyl-1,3,5-triazine-2,4,6(1H,3H,5H)-trione (**27**) [224,225]. Despite the general ease of this methodology, the majority of resulting hydroxide conductivities were relatively low, <10 mS/cm at room temperature. Finally, it should be noted that thiol reagents are generally malodorous, and there is a relatively limited commercial availability of polythiol reagents.

#### 5.2.6. Thermal Friedel–Crafts Electrophilic Substitution

Several articles have reported so-called ‘self-crosslinking’ of AAEM upon exposure to elevated temperatures after heat treating the membrane after solution casting. We speculate that, in these cases, it is likely that residual unconverted halide groups react via Friedel–Crafts electrophilic substitution, in aromatic polymers to form crosslinks, as shown in Figure 16, sometimes without the addition of a catalyst [152]. Friedel–Crafts substitution has been shown to be thermally initiated at temperatures as low as 80 °C, where the substitution can occur at adjacent aromatic rings [226]. Overall, this method presents a facile method to crosslink AAEMs without requiring additional chemicals/monomers or catalysts, which may mitigate the risk of contamination of hydrophilic domains compared to other methods [159]. Further, Xu et al. suggest an important benefit of Friedel–Crafts crosslinking might be the relatively little influence it has on the membrane’s ability to develop hydrophilic–hydrophobic phase separation at lower crosslink densities [78]. Despite the ease of crosslinking this method provides, quantifying and/or controlling the degree of crosslinking may be challenging, although control may possibly be exerted by adjusting the degree of halogenation or casting temperature.

#### 5.2.7. Epoxide Crosslinking

The polymerization of epoxide functional groups can be initiated in the presence of acid/base catalysts, or even simply at high temperatures (Figure 17). The opening of epoxide rings has shown to thermally initiate from 120–160 °C, low enough to avoid the onset of quaternary ammonium degradation. Kohl et al. utilized multifunctional epoxide tetraphenylolethane glycidyl ether (**28**) to crosslink polysulfone-based AAEMs and reported reduced WU, swelling, and methanol permeability relative to non-crosslinked systems [227]. Nijmeijer et al. crosslinked a polyvinyl alcohol-based AAEMs with polyethylene glycol diglycidyl ether (PEGDGE, **29**) and used XRD to identify the formation of new crystalline regions afforded by crosslinking of PEGDGE, which helped to counteract the competing reduction in crystallization due to new hydroxyl group formation [228]. In the crosslinking of 5-norbornene-2-methylene glycidyl ether (**30**) tethers in metathesis manufactured AAEM, FTIR was demonstrated as an effective tool for tracking epoxide polymerization [156]. Despite being a lesser-used methodology, epoxide crosslinking produces mechanically stable membranes with relatively effective hydroxide conductivities.

#### 5.2.8. Ionic Crosslinking

An additional crosslinking method involves the introduction of Lewis acid/base pairs to provide electrostatic, or ionic, interactions, as shown in Figure 18. This approach yields results analogous to those of covalent crosslinking methodologies. Electrostatic interactions lead to the formation of extended hydrogen bonding networks, which act to reinforce the membrane. Wu et al. developed a membrane formulation consisting of both imidazolium functionalized poly(ether ether ketone) (PEEK) and sulfonated PEEK with varying ratios of anion to cation groups; 1:17 to 3:17 sulfonated to imidazolium [229]. This work showed that increasing the amount of electrostatic interactions increased mechanical strength and enhanced stability but impeded ion mobility (up to 31.59 mS/cm at 30 °C). He and team demonstrated a similar ionic blending process to arrive at PEEK-based AAEMs with significantly improved mechanical strength (increased by roughly one order of magnitude) and reduced WU relative to uncrosslinked PEEK membranes [72]. Ren and co-workers used polysulfone functionalized with sulfonic acid and aliphatic amines to form ionic crosslinkages. In screening aliphatic chain lengths from 6 to 10, the authors report that eight carbon lengths promoted improved phase segregation and reported a near 50% increase in remaining IEC after stability testing and improved hydroxide conductivity [155]. Finally, nanomaterials, such as carbon nanotubes or graphene oxide (GO), have been incorporated to form AAEM composites with improved material properties. GO, for instance, can be functionalized to yield covalent crosslinks [230] or used as is to form likely ionic crosslinks between the carboxylic acids on the GO and ammonium groups in AAEM polymer [231]. 

#### 5.2.9. Cycloadditions

A few articles have reported cycloaddition reactions to crosslink AAEMs. Lee et al. demonstrated the effectiveness of (2+2+2) cyclotrimerization in polysulfone with alkyne end groups [70]. Trimerization occurred at 180 °C to form self-supporting matrix resistant to dissolution in NMP, with high hydroxide conductivities (107 mS/cm at 80 °C) and good alkaline stability. Liu et al. prepared crosslinked polysulfone AAEMs using (2+2) cycloaddition of trifluoro vinyl end groups resulting in the formation of perfluorocyclobutane crosslinks at 200 °C [76]. Ionic phase segregation was optimized with 5% per monomer crosslinking. IEC values correlated with corresponding feed values indicating that despite the high thermal treatment required to initiate crosslinking, quaternary amine retention was unaffected [76]. Hydroxide conductivities reported by both Lee and Lin exceeded 70 mS/cm at 80 °C, thereby demonstrating the utility of cycloadditions for producing crosslinked AAEMs [70,76].

#### 5.2.10. Radiation-Activated Crosslinking

Irradiation is an effective method to induce crosslinking in polymeric materials, through the formation of reactive radical species via gamma-induced or electron beam-induced C-H or C-C bond cleavage. This methodology has been used to functionalize PTFE and polyethylene (PE) scaffolds with VBC, likely with concomitant crosslinking of PTFE and PE chains [232]. Irradiation intensity and exposure time was used to control radical species generation for simultaneous quenching and covalent grafting onto polymer backbones. Films of PTFE and PE were first exposed to a prescribed dose, then immediately submerged into a solution containing VBC. Fang et al. crosslinked ETFE by first irradiating it with a dose of 90 kGy at a rate of 4 kGy h^−1^ in argon with a ^60^Co γ-ray source. The irradiated membrane was then immediately submerged in a 60 wt% solution of VBC in toluene to initiate grafting. Subsequent Menshutkin reaction was proceeded by adding DABCO followed by alkylation with p-Xylylenedichloride and TMA to form a crosslinked ETFE AAEM [232]. Little to no change in conductivity was reported after 120 h immersion in 60 °C 10 M KOH solution [233]. Alternatively, divinyl benzene has also been used to introduce crosslinks onto irradiated polymethylpenthene [145]. Espiritu et al. noted that radiation dosage also had a clear effect on AAEM stability. It was found that higher radiation intensities reduced the formation of oxidative degradation products that can ultimately reduce AAEM performance [234]. 

## 6. Conclusions and Perspective

In this review, we surveyed synthetic approaches for AAEM fabrication using crosslinking as a strategy to tune material properties. AAEMs are important components in several emergent electrochemical energy storage and conversion technologies, such as AEMFC and AEMWE. AAEM development continues at a rapid pace, and this is reflected in a wealth of peer-reviewed articles and the recent growth of commercial suppliers, including, but not limited to, Dioxide Materials (Boca Raton, FL, USA), Orion Polymers (Cohoes, NY, USA), Ionomr (Vancouver, BC, Canada), Versogen (Newark, DE, USA), Xergy (Harrington, DE, USA), Fumatech (Bietigheim-Bissingen, Germany), and Tokuyama (Tokyo, Japan). Crosslinking AAEMs increases network rigidity, thereby preventing excessive WU and enhancing dimensional stability. Alkaline stability is also generally enhanced in crosslinked AAEMs, which is likely a function of improved hydroxide solvation and increased steric hinderance around charge carrying, albeit hydroxide-label, cation groups. While crosslinking generally decreases ion conductivity relative to uncrosslinked analogs of similar IEC, the increased network rigidity supports higher IEC. Higher IEC is correlated with higher ion conductivity and in some cases, crosslinking enables the fabrication of AAEMs with higher conductivities relative to their uncrosslinked analogs. Finally, selectivity can also generally be tuned and/or improved through crosslinking.

Various synthetic approaches have been used to manufacture AAEMs with alkaline-resistant crosslinks. The most common approach for crosslinking AAEMs involves the addition of polyfunctional amines to displace pendant halides. Tertiary diamines, in particular, can directly form quaternary amine crosslinkages via Menshutkin-type reactions, thereby also enhancing IEC. Other approaches for crosslinking AAEMs include olefin methathesis, free-radical polymerization, ionizing radiation, thiol-ene and azide-alkyne click chemistries, etc. The wealth of alkaline-resistant crosslinking methodologies means that any AAEM polymer scaffold has amenable crosslinking, although it should be noted that there may be some drawbacks, particularly for certain processing/fabrication steps. Crosslinked materials are thermosets and are no longer soluble in solvents, which can complicate membrane casting. Crosslinks are therefore usually formed during or after casting. Further, highly crosslinked AAEMs uptake less water, which may hinder ion-exchange processes during the conversion of Cl^−^ for OH^−^, resulting in lower than theoretical IEC. 

Considerable challenges remain for both AAEM and AAEM device development. Alkaline resistance remains a major problem but one that has seen remarkable advances over the past 10 years with the development of sterically hindered cations, such as those based on poly(aryl piperidium) and poly(benzimidazolium) scaffolds. Additional AAEM fabrication challenges include the optimization of low-cost and low-toxicity materials, with a particular emphasis on transitioning away from perfluoroalkyl components/substances (PFAS) common to Nafion-type PEMs. The conversion of CO_2_ into commodity feedstocks (e.g., ethylene) in alkaline environments naturally mitigates the parasitic hydrogen evolution reaction but presents additional challenges for designing AAEMs. CO_2_ electrolysis produces a complex product mix, and certain liquid products such as alcohols and acids may pose additional degradation concerns, such as the potential dissolution of the membrane or formation more corrosive electrolytes (alkoxides > hydroxides). Finally, at the device level, electrolyte precipitation and carbonation (OH^−^ + CO_2_) pose significant challenges for commercial viability of AEMFCs, AEMWE, and CO_2_ electrolyzers. Crosslinking, in terms of both the nature of the crosslinker (i.e., hydrophobicity, rigidity, etc.) and extent or degree of crosslinking, is a useful tool that can optimize AAEM properties to meet the unique requirements and environments of various electrochemical devices. 

## Figures and Tables

**Figure 1 polymers-15-01534-f001:**
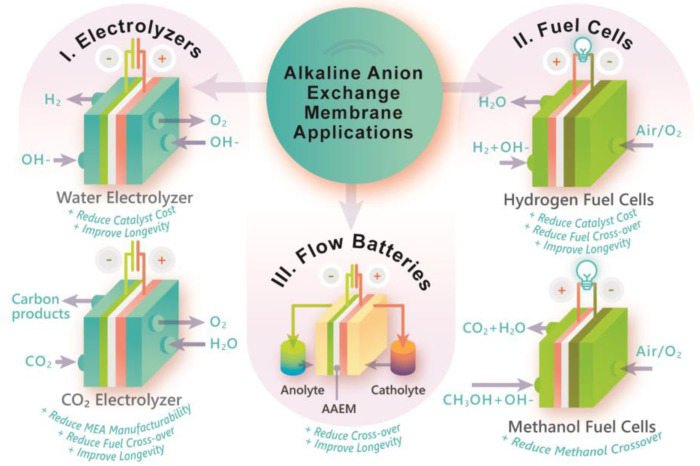
Depiction of the electrochemical system applications related to AAEMs and key shortcomings (cyan) that could improve by incorporating crosslinked AAEMs.

**Figure 2 polymers-15-01534-f002:**
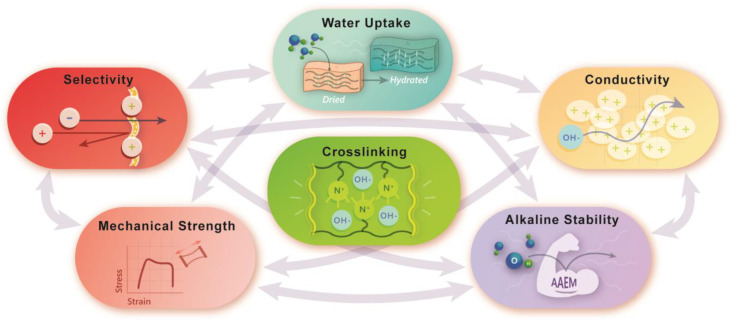
Graphical depiction of the interactions between important material parameters when considering crosslinking. Arrows between bubbles represent the interconnected relationship between the physical properties of AAEMs.

**Figure 3 polymers-15-01534-f003:**
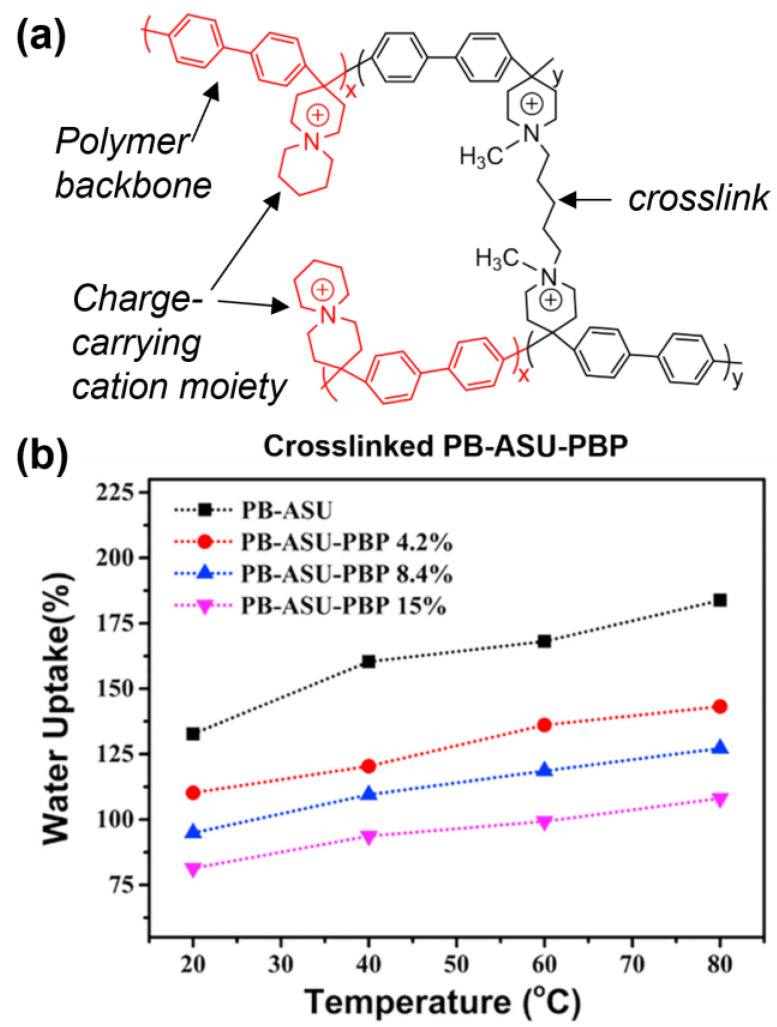
(**a**) Chemical structure of the controllable crosslinking of poly(arylene 6-azaspiro[5.5] undecanium). (**b**) Percent WU as a function of temperature. Adapted with permission from Ref. [81]. Copyright 2019 Elsevier.

**Figure 5 polymers-15-01534-f005:**
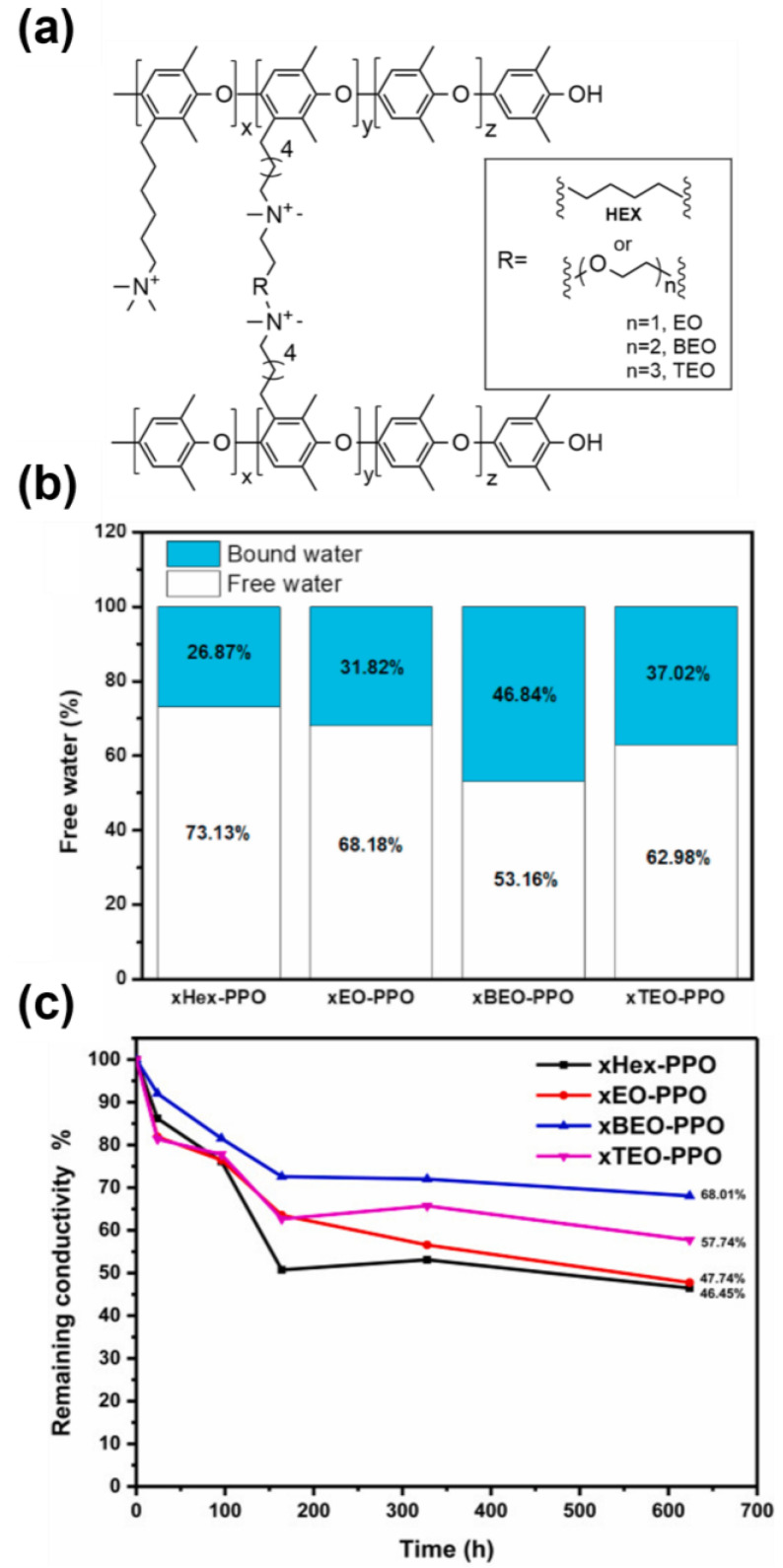
(**a**) Structure of the crosslinked PPO membranes with hexyl- and O-containing crosslinkers. (**b**) The relative fractions of free water and bound water. (**c**) Remaining hydroxide conductivity after the alkaline stability tests in a 1 M KOH solution at 80 °C. Adapted with permission from Ref. [86]. Copyright 2021 Elsevier.

**Figure 6 polymers-15-01534-f006:**
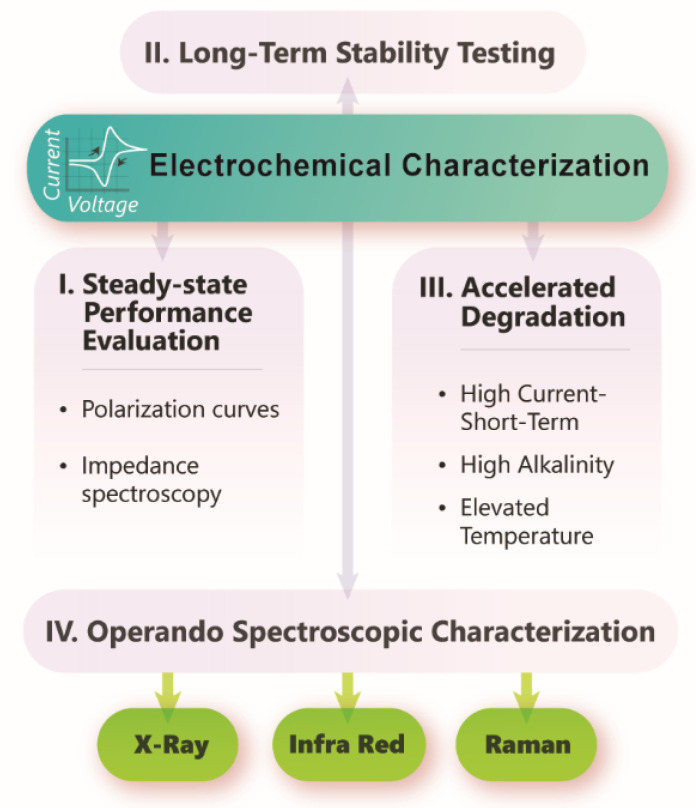
Aspects of electrochemical characterization techniques to be leveraged to improve the development and understanding of AAEMs.

**Figure 7 polymers-15-01534-f007:**
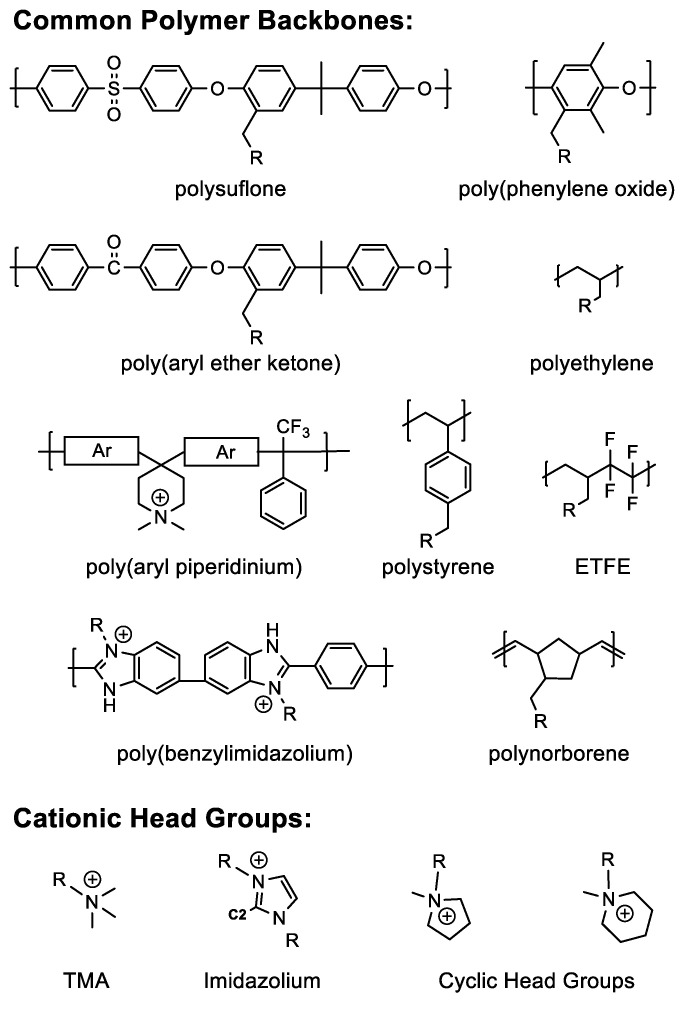
Common polymer backbones used in the synthesis of AAEMs as well as implemented cationic head groups. Abbreviations: TMA, trimethylammonium; ETFE, ethylene tetrafluoroethylene.

**Figure 8 polymers-15-01534-f008:**
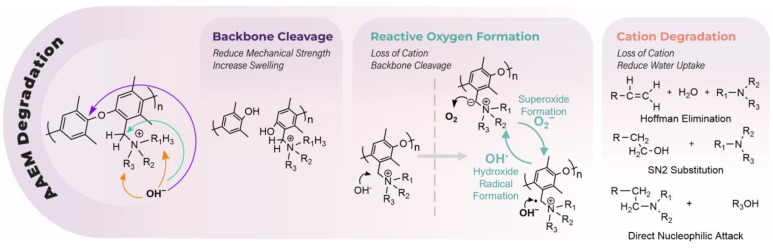
Primary degradation pathways of quaternary amine cation head groups.

**Figure 9 polymers-15-01534-f009:**
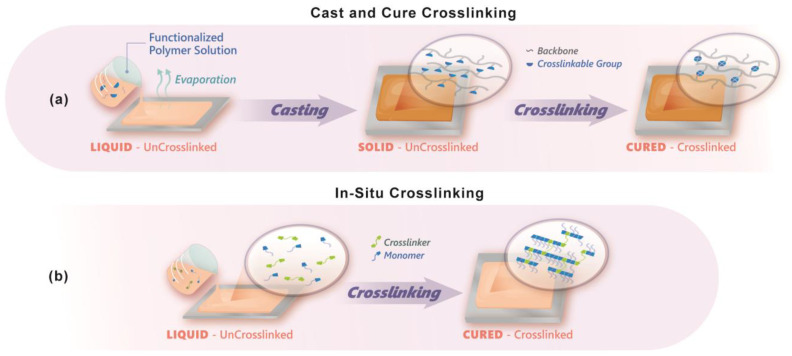
Illustration representing two methods of preparing crosslinked membranes by (**a**) cast and cure method or (**b**) in situ crosslinking.

**Figure 10 polymers-15-01534-f010:**
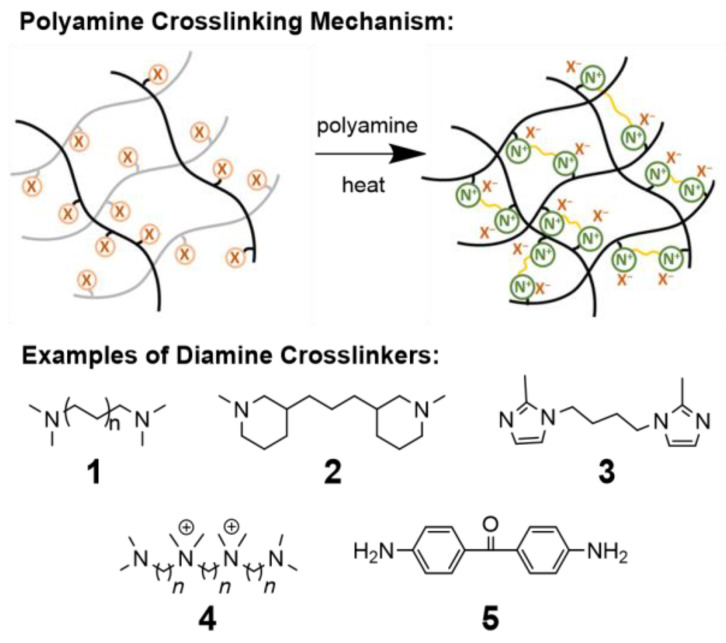
The Menshutkin reaction mechanism as well as diamine monomers used in AAEM crosslinking synthesis.

**Figure 11 polymers-15-01534-f011:**
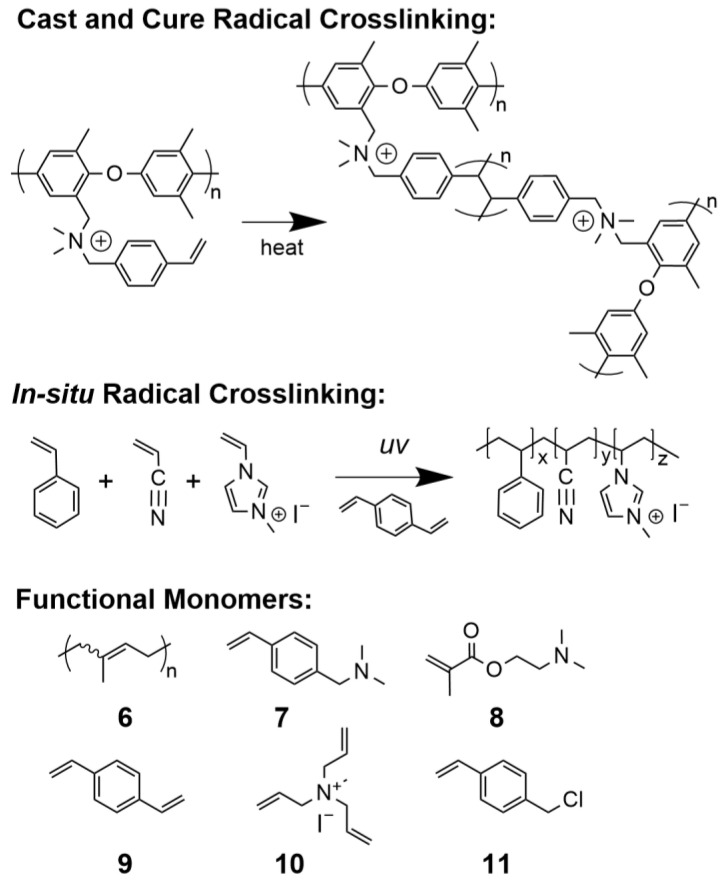
Free radical polymerization strategies for the fabrication of crosslinked AAEMs.

**Figure 12 polymers-15-01534-f012:**
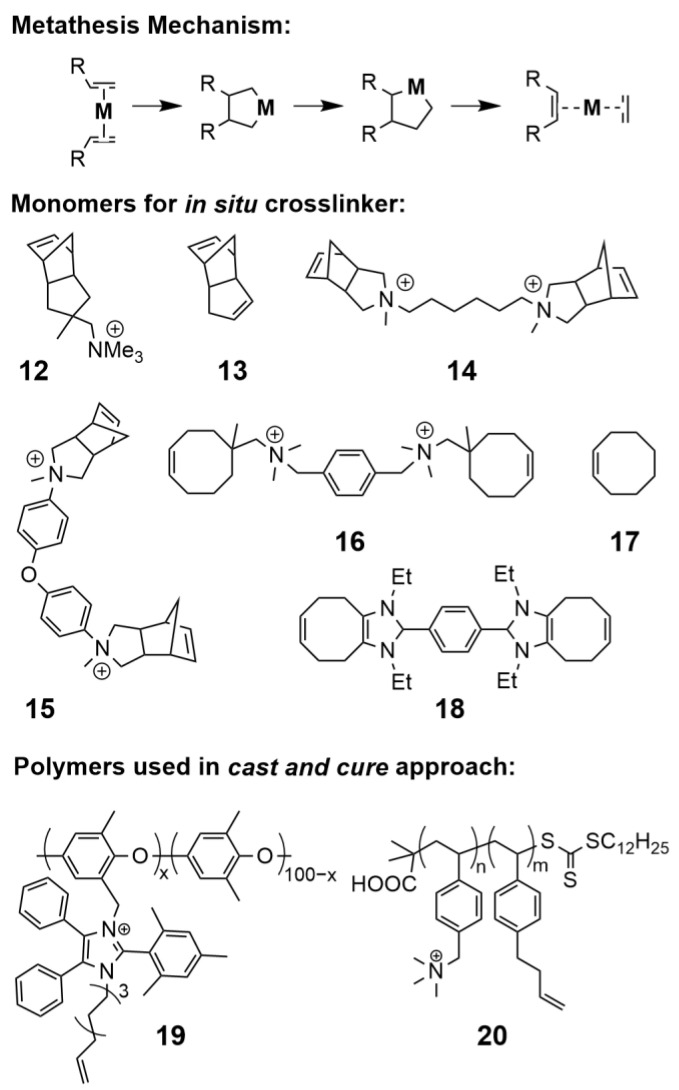
Metathesis crosslinking mechanism and corresponding monomers.

**Figure 13 polymers-15-01534-f013:**
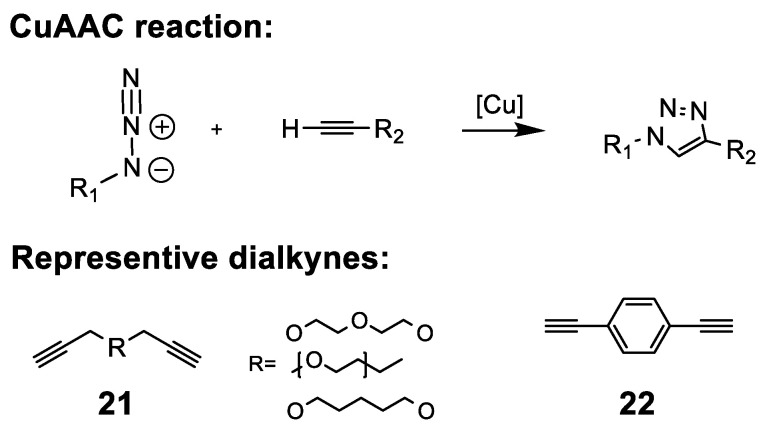
CuAAC crosslinking and functional monomers used.

**Figure 14 polymers-15-01534-f014:**
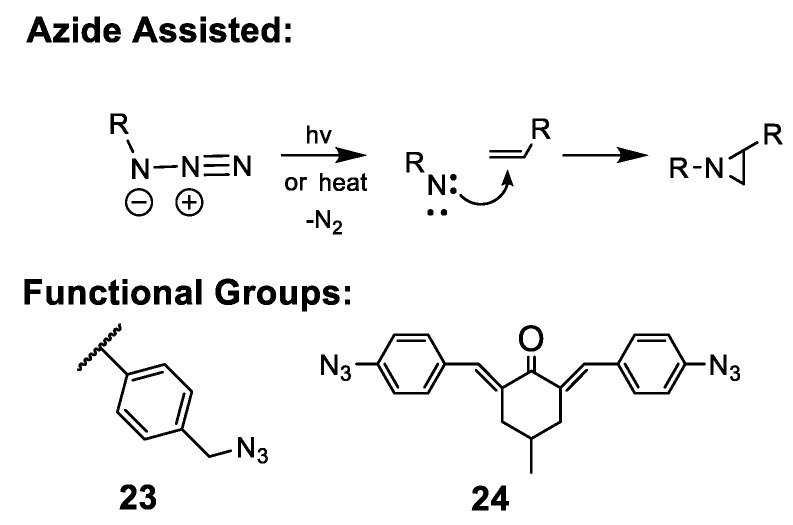
Azide-assisted crosslinking and functional monomers used.

**Figure 15 polymers-15-01534-f015:**
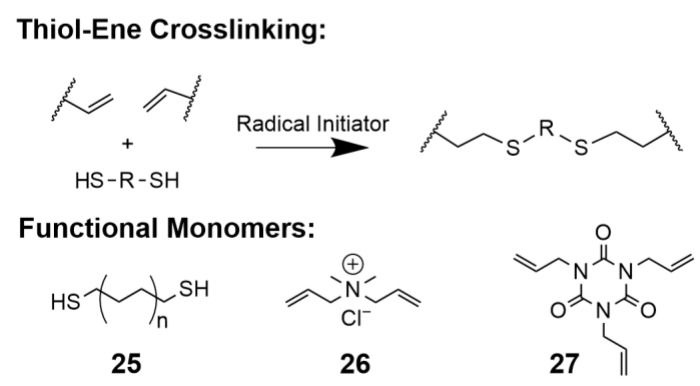
Thiol-Ene crosslinking and functional monomers used.

**Figure 16 polymers-15-01534-f016:**
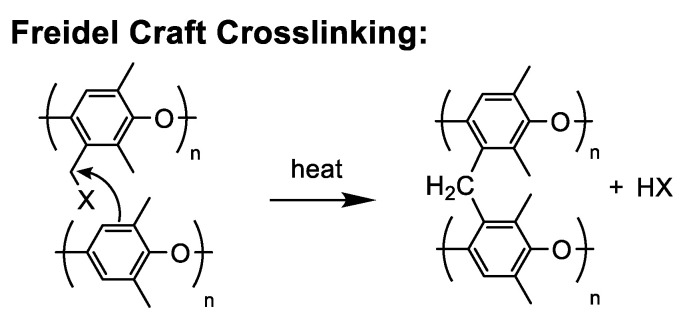
Friedel–Crafts crosslink mechanism.

**Figure 17 polymers-15-01534-f017:**
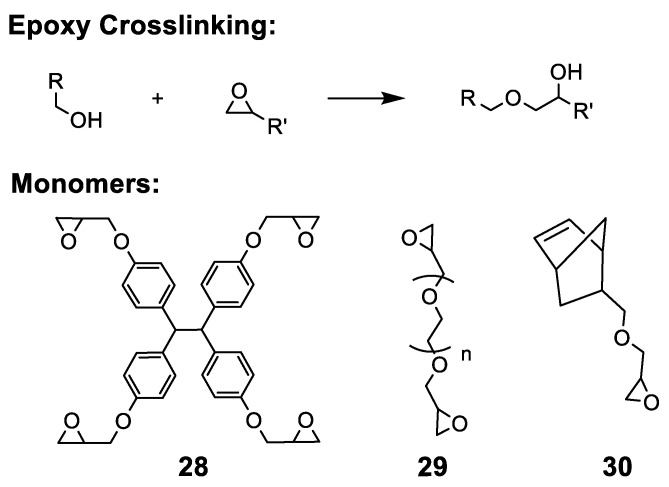
Epoxy esterification crosslinking mechanism and functional monomers.

**Figure 18 polymers-15-01534-f018:**
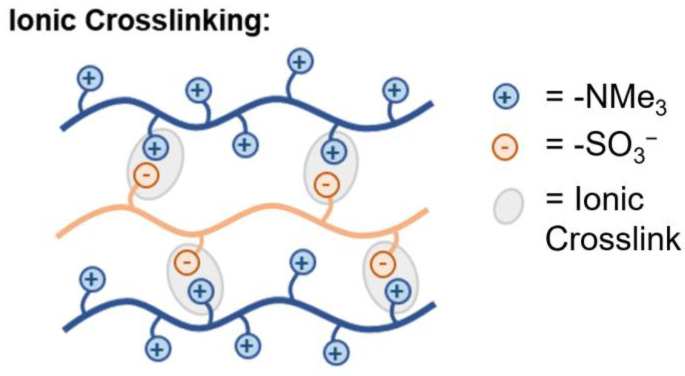
Illustrative depiction of blended ionic crosslinked AAEM.

## Data Availability

No new data were created or analyzed in this study. Data sharing is not applicable to this article.

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
