# Peer review of "Tuning Alkaline Anion Exchange Membranes through Crosslinking: A Review of Synthetic Strategies and Property Relationships"

_polymers, 2023, doi:10.3390/polym15061534_

Round 1

Reviewer 1 Report

This review is devoted to the systematization of the knowledge in the development of anion-exchange membranes, in particular, cross-linking as a method of tuning properties and characteristics. The division into chapters chosen by the authors is quite understandable and logical. The review is accompanied by good illustrations and reaction schemes and will interest readers of the Membranes. Therefore, I recommend this review for publication. Below is a list of comments that are advisory and can be taken into account at the discretion of the authors.

·        In the introduction, each application of anion-exchange membranes in Figure 1 is given a small paragraph with a brief description, except flow batteries.

·        As the authors note that water management is paramount for anion-exchange membranes. I have several references in mind where the effect of water content on conductivity - selectivity tradeoff for anion exchange membranes has been studied [10.1021/am403207w, 10.1002/app.48656, 10.1002/pol.20210304]. Perhaps the information in these links can be used to illustrate the relationship between conductivity and selectivity. Separately, I would like to note the work 10.1002/app.48656, in which, using the radiation-grafted membranes based on polystyrene, the influence of cross-linking on the change in properties is considered. In particular, there is mention of cross-linking of polystyrene both with divinylbenzene and at the stage of chloromethylation by the Friedel-Crafts mechanism, which can be viewed as another example of "self-crosslinking".

·        One promising application of anion exchange membranes is reverse electrodialysis—a process in which electrical energy is obtained from an electrolyte concentration gradient. Although RED uses anion-exchange membranes in tandem with cation-exchange membranes, optimizing the properties of anion-exchange membranes is one of the essential tasks in this area of research, and cross-linking is one of the critical tools [10.1039/d1ta05166k, 10.1016/j.memsci.2018.12.034]. Usually, a trade-off is considered between conductivity and perspectivity (analogous to selectivity). The only thing in this process is that anion-exchange membranes transfer chlorides, sulfates, nitrates, and not hydroxides. But since the authors consider vanadium flow batteries, where there is no transfer of hydroxides, they may feel it necessary to mention this application of anion-exchange membranes in their review as a particular case of flow batteries.

Minor remarks:

1) In figure 1, in the scheme of operation of the fuel cell, the arrow for hydrogen is indicated incorrectly.

2) what is "65" on line 716

Author Response

Reviewer Comment #1: In the introduction, each application of anion-exchange membranes in Figure 1 is given a small paragraph with a brief description, except flow batteries.

Author’s Response:  Thank you for your input and the authors agree. Two additional sentences briefly explaining RFBs are included in the introduction (paragraph 4) and reorganized for clarity.

Reviewer Comment #2: As the authors note that water management is paramount for anion-exchange membranes. I have several references in mind where the effect of water content on conductivity - selectivity tradeoff for anion exchange membranes has been studied [10.1021/am403207w, 10.1002/app.48656, 10.1002/pol.20210304]. Perhaps the information in these links can be used to illustrate the relationship between conductivity and selectivity. Separately, I would like to note the work 10.1002/app.48656, in which, using the radiation-grafted membranes based on polystyrene, the influence of cross-linking on the change in properties is considered. In particular, there is mention of cross-linking of polystyrene both with divinylbenzene and at the stage of chloromethylation by the Friedel-Crafts mechanism, which can be viewed as another example of "self-crosslinking".

Author’s Response: Thank you for the additional citations. These articles will surely help illustrate the relationship between crosslinking and selectivity. They have been added to section 3.4 with accompanied discussion. Additionally, 10.1002/app.48656 was added to section 5.2.10 as an example of radiation grafted crosslinking.

Reviewer Comment #3: One promising application of anion exchange membranes is reverse electrodialysis—a process in which electrical energy is obtained from an electrolyte concentration gradient. Although RED uses anion-exchange membranes in tandem with cation-exchange membranes, optimizing the properties of anion-exchange membranes is one of the essential tasks in this area of research, and cross-linking is one of the critical tools [10.1039/d1ta05166k, 10.1016/j.memsci.2018.12.034]. Usually, a trade-off is considered between conductivity and perspectivity (analogous to selectivity). The only thing in this process is that anion-exchange membranes transfer chlorides, sulfates, nitrates, and not hydroxides. But since the authors consider vanadium flow batteries, where there is no transfer of hydroxides, they may feel it necessary to mention this application of anion-exchange membranes in their review as a particular case of flow batteries.

Author’s Response: RED devices require membrane designers to deliberately balance conductivity and selectivity and we believe that crosslinking is a valuable tool for this design task. We have included the suggested references within section 3.4 as examples of other AAEM-based dialysis applications that benefit from crosslinking.

Reviewer Comment #4: In figure 1, in the scheme of operation of the fuel cell, the arrow for hydrogen is indicated incorrectly.

Author’s Response: Thank you for the correction. The arrows for the hydrogen and oxygen in fuel cell and methanol fuel cell have been corrected.

Reviewer Comment #5: what is "65" on line 716

Author’s Response: The unbracketed 65 was a mis-formatted citation and has since been corrected. 

Reviewer 2 Report

The authors have reviewed synthetic methodologies for incorporating crosslinks during anion exchange membrane (AEM) fabrication and highlight necessary precautions for each approach. Overall, the article is interesting, but a few major issues need to be addressed before being published in “Polymers” journal. This article can be further improved by addressing the following suggestions:

1. The introduction section should cover recent developments in the fabrication of AEM for fuel cells, water electrolyzers, flow batteries, and CO2 electrolyzers.

2. The authors should also discuss the drawbacks and challenges of using crosslinked anion exchange membranes in water electrolyzer and fuel cell applications.

3. This study did not fully disclose the AEM's operational decline mechanism. This should be stated in the introduction section.

4. Despite the greater mechanical and alkaline stability, the cross-linking approach has been found to affect the hydroxide conductivity of AEM in some previous researches. The authors need to review that material and report what they discover.

5. To contrast the electrochemical performance of the newly fabricated cross-linked AEMs, the authors must draw a comparison table.

6.  In the introduction or discussion section, some closely related recent sources can be cited, such as: doi.org/10.1016/j.ijhydene.2022.10.184;

7. The future challenges of efficient AEM fabrication should be written in the conclusion part of the review.

8. There are several grammatical and writing mistakes in the review. The English of the article may need some improvement.

Author Response

Reviewer Comment #1: The introduction section should cover recent developments in the fabrication of AEM for fuel cells, water electrolyzers, flow batteries, and CO2 electrolyzers.

Author’s Response: We have expanded the Introduction to include additional recent developments in AAEM research, including backbone bound cations (to mitigate hydroxyl attack by increasing steric hinderance) and the transition to pure hydrocarbon thermoplastics (as a means to reduce backbone degradation). These recent developments are featured in commercial AAEM products (see new text, lines 113-123). Crosslinking within Sustainion and Xergy Inc. AAEMs are also discussed within the context of the manuscript.

Reviewer Comment #2: The authors should also discuss the drawbacks and challenges of using crosslinked anion exchange membranes in water electrolyzer and fuel cell applications.

Author’s Response: We thank the Reviewer for noting the importance of reviewing the nontrivial drawbacks and challenges introduced by crosslinking. We have expanded this discussion from our initial draft, which briefly addresses crosslinking drawbacks in the 1st paragraph of Section 2. Transitioning from a thermoplastic to an irreversible thermoset is one major challenge which we have expanded upon. Additional emphasis on drawbacks and challenges has been included, along with a discussion of limitations on post-performance characterization due to crosslinking and additional synthesis and fabrication costs that result from adding crosslinking chemistry as a processing step. The most commonly cited drawback, reduced conductivity (hindered ion transport), is detailed in section 3.2 and now prefaced in the introduction and mentioned in the conclusion. Additional citations from within the original text were added to section 3.2 (lines 288-230) as examples of AAEMs with reduced conductivity as a result of crosslinking.

Reviewer Comment #3: This study did not fully disclose the AEM’s operational decline mechanism. This should be stated in the introduction section.

Author’s Response: Additional text is added to the introduction (lines 98-109) highlighting the simplified mechanisms for device degradation and failure, which is largely dependent on the forementioned membrane degradation modes and discussed in more detail in section 5.1.

Reviewer Comment #4: Despite the greater mechanical and alkaline stability, the cross-linking approach has been found to affect the hydroxide conductivity of AEM in some previous researches. The authors need to review that material and report what they discover.

Author’s Response: We expanded our discussion of conductivity in section 3.2 and incorporated a new figure (Figure 4) to highlight some of the nuances associated with crosslinking and conductivity. With the additional clarity from the edits implemented from Reviewer 2 Comment #2, we believe that the commonly perceived tradeoff is now captured. All other factors being equal (IEC, water uptake, etc.), a crosslinked membrane would generally show lower conductivity compared to its uncrosslinked analog, however this relationship is far from simple. The increased network rigidity and lower WU provided by crosslinking supports higher IEC. Higher IEC is correlated with higher ion conductivity and in some case, crosslinking enables fabrication of AAEMs with higher conductivities relative to their uncrosslinked analogs. This is especially once other properties such as dimensional stability and mechanical strength are taken into account

Reviewer Comment #5: To contrast the electrochemical performance of the newly fabricated cross-linked AEMs, the authors must draw a comparison table.

Author’s Response: We agree with this comment and thank Reviewer #2 for motivating us to develop the newly added Figure, Figure 4, which shows a comparison of conductivity and dimensional stability for AAEM studies that compared properties of crosslinked to uncrosslinked membranes. In section 3, the leading paragraph prefaces comparisons and conclusions that we draw from published work.

Reviewer Comment #6: In the introduction or discussion section, some closely related recent sources can be cited, such as: doi.org/10.1016/j.ijhydene.2022.10.184

Author’s Response: We thank reviewer #2 for this reference, very interesting work, and we have seen an uptick in articles reporting AAEMs that incorporate graphene oxide and other nano-scale fillers. We have added this reference to section 5.2.8 “Ionic Crosslinking” along with an accompanying discussion.

Reviewer Comment #7: The future challenges of efficient AEM fabrication should be written in the conclusion part of the review.

Author’s Response: Thank you for your comment. The Conclusions have been restructured to include future challenges in AAEM fabrication which include developing alkaline stable AAEMs that outlast Nafion, developing cheaper AAEMs, manufacturing AAEMs without fluorinated components, and understanding and designing AAEMs for complicated reactor systems such as CO2 electrolysis.

Reviewer Comment #8: There are several grammatical and writing mistakes in the review. The English of the article may need some improvement.

Author’s Response: Thank you for your concern. Additional edits were made to fix remaining grammatical errors.

Round 2

Reviewer 2 Report

This paper can be acceptable in its present form as the manuscript has been revised. However, the authors should again carefully check for typos and reference formatting.